# Psychometric properties of the Adult Primary Care Assessment Tool Short form (PCAT-S) among high-risk patients in Australian general practice

Chau M. Bui[1]*, Marijka J. Batterham[1,2], Judy Mullan[1], Gregory Peterson[3], Christine Metusela[1], Jan Radford[4], Simon Eckermann[5], Danielle Mazza[6], Grant Russell[6], Andrew Bonney[1]

1 Graduate School of Medicine, Faculty of Science, Medicine and Health, University of Wollongong, New South Wales, Australia, 2 Statistical Consulting, Engineering and Information Sciences, University of Wollongong, New South Wales, Australia, 3 School of Pharmacy and Pharmacology, University of Tasmania, Sandy Bay, Australia, 4 Launceston Clinical School, Tasmanian School of Medicine, University of Tasmania, Launceston, Australia, 5 School of Health and Society, University of Wollongong, Wollongong, New South Wales, Australia, 6 Department of General Practice, School of Public Health and Preventive Medicine, Monash University, Victoria, Australia

* cbui@uow.edu.au

## Abstract

### Introduction

The Primary Care Assessment Tool (PCAT) is designed to assess a patient's experience with primary care across various core and ancillary domains, including First contact – Utilization, First contact – Access, Ongoing Care, Coordination, Comprehensiveness (services provided), Family-centeredness, Community Orientation, and Cultural Competence. This study examined the psychometric properties of the Adult Primary Care Assessment Tool Short Form (PCAT-S) in the Australian general practice setting.

### Method

Data included 715 participants from the EQuIP-GP study, a cluster randomized controlled trial (RCT) conducted with adults aged 18–65 years with a chronic illness or aged over 65 years, from 34 general practices across Australia. For each subscale we assessed internal consistency using Cronbach's alpha. Factor structure of the PCAT-S instrument was assessed through confirmatory and exploratory factor analysis, using three samples with different methods for handling 'don't know/can't remember' responses.

**Data availability statement:** Requests for data may be made to the authors, or if unavailable, through the UOW Health and Medical Human Research Ethics Committee (uow-humanethics@uow.edu.au).

**Funding:** This study used funding from a larger project funded by the Australian Government Department of Health and Aged Care (https://www.health.gov.au/). This funding was provided via the Royal Australian College of General Practitioners (RACGP, https://www.racgp.org.au/). The funding bodies had no role in the design, data collection and analysis, decision to publish, or preparation of the manuscript. GR, JR and DM have received honoraria from the RACGP for expert committee roles. The other authors have no conflicts to declare.

**Competing interests:** I have read the journal's policy and the authors of this manuscript have the following competing interests: Prof Grant Russell, A/Prof Jan Radford, and Prof Danielle Mazza have received honoraria from the RACGP for expert committee roles. The other authors have no conflicts to declare. This does not alter our adherence to PLOS ONE policies on sharing data and materials. Requests for data may be made to the authors, or if unavailable, through the UOW Health and Medical Human Research Ethics Committee (uow-humanethics@uow.edu.au).

## Results

The findings were mixed. Specifically, the subscales related to First Contact – Utilization, Ongoing Care and Comprehensiveness, demonstrated satisfactory internal consistency. However, the remaining subscales showed weak internal consistency. Confirmatory factor analysis indicated potential model misspecification, while exploratory factor analysis generally supported the hypothesized factor structure, albeit with some observed deviations.

## Conclusions

The findings indicate the PCAT-S shows promise as an instrument to evaluate primary care experiences in Australia. However, the observed variability in internal consistency, along with issues identified in confirmatory and exploratory factor analyses, highlight the need for further validation and refinement in this population. Further research is required to address the identified limitations and enhance the tool's applicability within the Australian general practice context.

## Introduction

Primary health care is widely recognized as fundamental to health systems for attaining universal health coverage, better health outcomes, and health equity [1,2]. Many countries have prioritized strengthening primary care as part of health system reforms [2]. Globally, there has been a growing emphasis on measuring patient experiences as a means to assess health system performance [3].

The Primary Care Assessment Tool (PCAT) was developed by the Johns Hopkins Primary Care Policy Centre [4]. The PCAT is designed to assess the extent and quality of primary care services in provider settings identified by consumers as their predominant source of health care.

The PCAT is based on a theoretical model of primary care formulated by Starfield [5], which incorporates the following four essential primary care attributes: first-contact accessibility and use, continuity, comprehensiveness, and coordination, as well as family-centeredness, community orientation, and cultural competence.

The development of the PCAT began with the Children and Adolescents version, which was initially validated in the United States by Cassady et al. (2000) [6]. This was followed by the development and validation of the Adult Edition and the Adult Short Form, with further validation and refinement documented by Shi et al. (2001) [7].

The PCAT has since been translated, validated, and utilized across many countries, including Canada, Brazil, Malta, Spain, South Korea, Japan, China, Hong Kong, Taiwan, Tibet, Vietnam, Malawi, Uganda, and South Africa [4–8]. As noted by Rocha and colleagues [8], while many tools assess specific dimensions of primary care quality, there are few like the PCAT that enable a comprehensive evaluation of primary care from the population perspective. Additionally, the Consumer/Client Surveys are designed for self-administration and require only a high school reading level, making the tool broadly accessible to the general population [7].

To date, there have been no studies investigating psychometric properties of the PCAT when applied in Australia. General practice plays a central role in Australia's primary healthcare system, with general practitioners (GPs) typically serving as the first point of contact in the health system for an individual seeking health care [9]. While there are several tools which have been validated for measuring Australian patient experiences in general practice settings, such as the Doctors' Interpersonal Skills Questionnaire (DISQ) [10], the Practice Accreditation and Improvement Survey (PAIS) [11] and the Patient Enablement and Satisfaction Survey (PESS) [12], these tools are not designed to provide a holistic evaluation of primary care. In contrast, the PCAT is a multidimensional scale developed to assess several key structural and process features of primary care, including four core (first contact care, person-focused care over time, comprehensiveness, and coordination) and three ancillary (family orientation, community orientation, and cultural competence) broad theoretical concepts of primary care [4,6].

Investigating the psychometric properties of the PCAT in Australia presents an opportunity to provide a comprehensive measure of patient experiences with primary care, facilitating international comparisons. Undertaking comparative analyses can offer valuable insights into the relative strengths and weaknesses of primary care across different countries. This study therefore aimed to evaluate the psychometric properties of the 28-item abridged version of the adult patient survey, referred to as the Primary Care Assessment Tool Short Form (PCAT-S), among patients aged 18–65 years with a chronic illness or aged over 65 years, in general practice settings in Australia. The aim of this study was to examine the internal consistency and factor structure of the PCAT-S.

## Materials and methods

### Participants and procedures

We used data from a clustered two-arm Randomised Control Trial (RCT) which ran from 1 August 2018 to 31 July 2019, evaluating an intervention involving patient enrolment and a funding model for higher-risk patients in metropolitan, regional, and rural Australia [13]. Details of patient recruitment and survey administration have been published elsewhere [13–15]. Briefly, the RCT recruited 774 adult participants from 34 general practices. Recruitment of practices was between 1 April 2018 and 31 August 2018. Recruitment of eligible patients of consenting practices was between 1 May 2018 and 31 December 2018. To be eligible, adult participants needed to have attended the practice three or more times in the last 2 years, and be either: (i) aged 18–65 years with a chronic condition (such as chronic obstructive pulmonary disease, diabetes, ischaemic heart disease, cardiac failure, or asthma), or (ii) aged over 65 years.

Ethics approval was provided by the University of Wollongong Human Research Ethics Committee, Monash University and the University of Tasmania (2017/417). The trial was registered on the Australian New Zealand Clinical Trials Registry (ACTRN12618000105246). All participants provided written informed consent.

The PCAT-S was included in a broader questionnaire administered to the study participants at trial entry and trial completion, either online or through paper-based self-completion or by telephone interview. The questionnaire also included questions on the patient's socio-demographic background, health service utilisation within the past 12 months, and overall health-related quality of life measured using the EQ-5D-5L instrument [16,17]. To identify each participant's primary care provider (PCP), the survey employed the same three questions and algorithm as the original PCAT: (i) *Is there a doctor to whom you usually go if you are sick or need advice about your health? (usual source)*, (ii) *Is there a doctor who knows you best as a person? (knows best)* and (iii) *Is there a doctor who is most responsible for your health care? (most responsible)* [7]. Similar to the original PCAT, participants needed to have answered 'yes' to the question '*Have you ever had a visit to any kind of specialist or special service (like a surgeon or kidney specialist)'* to be eligible to respond to items in the Coordination subscale. A small number of adjustments were made to the Adult PCAT-S provided by tool developers and are listed in Supplementary S1 Table. Note the alphabetical ordering of the subscales from the original PCAT-S has not been retained due to the exclusion of two subscales—Coordination (Information Systems) and Comprehensiveness (Services Available) (see Supplementary S1 Table).

## Data analysis

This analysis focused only on the 28 items used to assess primary care attributes, organised into eight subscales: First contact – Utilization, First contact – Access, Ongoing Care, Coordination, Comprehensiveness (services provided), Family-centeredness, Community Orientation and Culturally Competent. Each subscale comprises 2–5 items. Response options for each item are measured on a 4-point Likert-type scale (1 = definitely not; 2 = probably not; 3 = probably yes; and 4 = definitely yes), with an additional response option 'don't know/can't remember', which were treated as missing data. All items are worded in a positive direction whereby higher scores equate to better experiences.

For this analysis, we used data from 715 adult participants who responded to any item/s of the PCAT-S at entry into the trial. Baseline responses from participants in both control and intervention arms were used for assessing factor structure and internal validity of the PCAT-S.

We report patient socio-demographic characteristics as well as an assessment of the data quality (proportion of missing data and 'don't know/can't remember' responses). Large ceiling or floor effects were defined as items where > 20% of respondents scored the highest (ceiling) or lowest (floor) possible score.

All analyses used R version 4.3.1 [18]. We used the package *psych* version 2.4.3 [19] to conduct sample adequacy tests, exploratory factor analyses and calculation of Cronbach's alpha, and *lavaan* version 0.6.16 [20].

(i)  Factor structure

*Factor analysis* examines the relationships between survey items to determine if responses from different subsets of items relate more closely to each other than to other subsets [21]. Confirmatory factor analysis (CFA) is used to confirm an existing theoretical model, or if the data do not fit the theoretical model, exploratory factor analysis (EFA) can be used to identify plausible underlying constructs for a set of items [21]. In the original PCAT validation study (conducted in two distinct subpopulations in the United States) Family-Centeredness emerged as a distinct factor only in the predominantly white, higher-income sample, while Cultural Competence emerged as a distinct factor only in the predominantly non-white, lower-income sample [7]. In several country-specific validation studies, factor analytic models did not exactly support the theoretical factor structure [22–26]. These examples highlight the need to validate the PCAT's factor structure for different health system contexts.

We conducted the Kaiser-Meyer-Olkin (KMO) statistic and Bartlett's Test for Sphericity to evaluate whether our sample was appropriate for carrying out factor analysis. We considered a KMO > 0.60 and significance of Bartlett's test of sphericity value (p < 0.05) to indicate suitability [27,28]. Multicollinearity risk was assessed by visualising correlation matrix and examining for small determinant of the correlation matrix (< 0.00001) – results are and is presented in S2 Table [29].

Factor structure was examined using confirmatory factor analysis (CFA) with a diagonally weighted least squares (DWLS) estimator, as is appropriate for ordinal data [30]. The 28 items of the PCAT-S were specified to load onto the eight factors as per the original PCAT, with each item loading only on its respective factor. We determined model fit using standard criterion and accepted benchmarks: Comparative Fit Index (CFI ≥ 0.90), Standardized Root Mean square Residual (SRMR <0.08), and Root Mean Square Error of Approximation (RMSEA ≤0.06) [31–33]. We reported scaled statistics, which corrects for the sample statistics of latent response variables being estimated with less precision than for observed variables [34]. We considered factor loadings of above 0.5 as acceptable.

Since the data did not fit the hypothesised CFA model [7], additional exploratory factor analysis was conducted. When a hypothesized CFA model does not adequately fit the data, it is acceptable practice to use EFA to explore the underlying factor structure and identify alternative item loadings or configurations [35,36]. To confirm the number of factors, we examined a Cattell's scree plot of eigenvalues. Response data were treated as continuous. We first used an oblique (oblimin) factor rotation method which assumes factors are non-independent. We assessed the correlation matrix describing relationships between factors to determine if correlations were low (defined as *r* < |0.30|) (S3–S5 Tables). Since we found most correlations to be low, we then used an orthogonal (varimax) factor rotation method which assumes factors are

independent and not correlated. Items were considered loaded onto a factor if the factor loading was > 0.40 for that factor and <0.40 for all other factors [37].

Both exploratory and confirmatory factor analysis require samples with no missing data, meaning data could only be used from participants who responded to all 28 items with a valid response (either '1 = definitely not', '2 = probably not', '3 = probably yes', or '4 = definitely yes'). These are called 'complete observations'. Only 180 (of 715) participants had complete observations, which was too small for a meaningful factor analysis. Consequently, we used two data imputation methods to handle 'don't know/can't remember' responses and conducted separate analyses for each method. The first method, recommended by PCAT developers, involved recoding 'don't know/can't remember' responses to '2 (probably not)' for items within a subscale where participants had valid responses to at least 50% of the items in that subscale. This method (applicable for all subscales excluding Comprehensiveness) interprets a patient's uncertainty about an item as indicating a negative perception of the service [4]. Using this 'developer-recommended' imputation method, after excluding individuals with any missing responses, there were 373 observations available for factor analysis.

The second method involved recoding all 'don't know/can't remember' responses to a neutral value, regardless of how many items within the subscale had valid responses— a method used in other PCAT studies [8,22,24,38]. Using this 'neutral value' imputation method, after excluding individuals with any missing response, there were 606 observations available for factor analysis.

(ii)  Internal consistency

*Internal consistency* evaluates the extent to which all the items in a test measure the same concept or construct [39]. In cultural adaptations of the PCAT, internal consistency was generally reported to be in an acceptable range [38,40,41]. However, in the original PCAT validation study, a lower than acceptable score was observed only for one subscale, First contact – Utilization (α = 0.64) [7]. In a subsequent Canadian study, the PCAT-S was found to have lower than acceptable internal consistency scores for First contact – Utilization (α = 0.68) and Community Orientation (α = 0.65), but acceptable scores for the remaining subscales (range 0.72–0.76) [26].

The internal consistency of each subscale of the PCAT-S was evaluated using Cronbach's alpha. We considered a Cronbach's alpha of >0.70 to indicate a minimally reliable subscale. Missing values were handled through pairwise deletion.

## Results

Patient and health service utilisation characteristics of all 715 respondents who responded to any item are presented in Table 1. Most participants were female (60%), born in Australia (83%) and spoke English at home (99%). The mean age of the sample was 66.9 years. Around 50% resided in a higher socioeconomic area, and most participants owned or mortgaged their own home (82%). Over half (62%) of participants were retirees, and just under a quarter (23.8%) were employed in some form. Over one-quarter of participants (26.9%) held a vocational or university qualification. More than half of the participants had private health care coverage for the full 12 months prior to completing the survey (65.5%), and the majority (76.8%) had a relationship with their GP or practice for over five years.

### Data quality

We present a summary of participant responses in Table 2. Overall, the item response rate was high. With the exception of items in the Coordination subscale, the proportion of missing responses was < 2% for each item. Most participants (n = 651, 91%) reported having visited a specialist or special service and were eligible to provide responses to items in the Coordination subscale. The proportions of the response category 'not sure/don't remember' ranged widely from 0.1 to 37.8%. Particularly high proportions were found in all items of the Community Orientation subscale (I1, I2, and I3), as well as the last two items in the First Contact – Access subscale (D3 and D4), with over 20% of responses falling into this

**Table 1. Demographic and health characteristics of total study participants (N = 715).**

| Characteristic | Frequency (%) |
|---|---|
| ***Socio-demographic*** | |
| Sex | |
| Male | 284 (39.7) |
| Female | 431 (60.3) |
| Age (years) | |
| 18–64 years | 263 (36.8) |
| 65 years and older | 451 (63.2) |
| Mean age (standard deviation) | 66.9 years (SD 13.5) |
| Socioeconomic status | |
| 5th quintile – least disadvantaged | 144 (20.1) |
| 4th quintile | 215 (30.1) |
| 3rd quintile | 167 (23.4) |
| 2nd quintile | 58 (8.1) |
| 1st quintile – most disadvantaged | 131 (18.3) |
| Primary language spoken at home | |
| English | 706 (98.7) |
| Other language | 7 (1) |
| Born in Australia | |
| Yes | 596 (83.4) |
| No | 118 (16.5) |
| Accommodation | |
| Owner occupied/ mortgaged | 587 (82.1) |
| Rented from a private landlord | 56 (7.8) |
| Other arrangements | 50 (7) |
| Rented from the Department of Housing | 22 (3.1) |
| Employment | |
| Retired from paid work | 443 (62) |
| Employed | 170 (23.8) |
| Unable to work (long-term sickness or disability) | 53 (7.4) |
| Looking after home/family | 27 (3.8) |
| At school or in full-time education | 9 (1.3) |
| Unemployed and looking for work | 9 (1.3) |
| Highest level of education attained | |
| Bachelor degree or above | 192 (26.9) |
| Diploma or TAFE/trade certificate | 222 (31) |
| Higher school certificate (Year 12) | 203 (28.4) |
| School certificate (Year 10) | 24 (3.4) |
| Primary school | 29 (4.1) |
| No qualification | 39 (5.5) |
| Weekly household income | |
| $2,500 or more | 63 (8.8) |
| $1,500—$2,499 | 62 (8.7) |
| $1,000—$1,499 | 151 (21.1) |
| $600—$999 | 141 (19.7) |
| $300—$599 | 167 (23.4) |
| Under $300 | 32 (4.5) |

*(Continued)*

**Table 1.** (Continued)

| Characteristic | Frequency (%) |
| --- | --- |
| Duration of relationship with GP or practice | |
| Over 5 years | 549 (76.8) |
| Between 3–4 years | 71 (9.9) |
| Between 1–2 years | 51 (7.1) |
| Less than a year | 23 (3.2) |
| *Health status* | |
| Moderate, severe or extreme problems with: | |
| Mobility | 320 (44.8) |
| Personal care | 31 (4.3) |
| Usual activities | 130 (18.2) |
| Pain or discomfort | 270 (37.8) |
| Anxiety or depression | 125 (17.5) |
| Health rating (0 = worst health, 100 = best health) | |
| Median EQ VAS rating (inter-quartile range) | Median 75.0 (IQR 29) |
| *Health services utilisation in the past 12 months (%)* | |
| Private health care coverage within the past 12 months | |
| All year | 468 (65.5) |
| Part of the year | 19 (2.7) |
| None | 217 (30.3) |
| Concession card use in the past 12 months | |
| Yes | 464 (64.9) |
| No | 251 (35.1) |
| Number of emergency department visits in past 12 months | |
| None | 470 (65.7) |
| One or two visits | 191 (26.7) |
| Three or more visits | 45 (6.3) |
| Number of overnight hospital visits in past 12 months | |
| None | 504 (70.5) |
| One or two visits | 171 (23.9) |
| Three or more visits | 32 (4.5) |
| *Among patients who had at least one hospital visit in the past 12 months (N = 203)* | |
| Saw their GP or practice within one week of the hospital visit | |
| Yes | 133 (65.5) |
| No | 64 (31.5) |
| Number of nights spent in hospital in the past 12 months | |
| Less than 10 days | 155 (76.4) |
| More than 10 days | 48 (23.6) |

For certain characteristics, percentages may not sum to 100%. We did not report frequencies where characteristics were either inadequately described, unknown or missing, or too few in number.

category. All subscales demonstrated the full range of possible scores. Based on the standard 20% criterion, 5 of the 28 items showed large floor effects and 23 items showed large ceiling effects. Additionally, very high ceiling effects (>80%) were noted for the two first-contact utilization items (C1 and C2) and for item J1.

**Table 2. Descriptive statistics of PCAT-S items from sample (N = 715).**

| Subscale and item | Mean (SD) | Floor/ Ceiling effect | Frequency (%) | |
|---|---|---|---|---|
| | | | True missing (%) | Don't know/ can't remember (%) |
| **First contact – Utilization** | | | | |
| C1. When you need a regular general checkup, do you go to your PCP before going somewhere else? | 3.95 (0.26) | 0.3/ 95.4 | 1 (0.1) | 2 (0.3) |
| C2. When you have a new health problem, do you go to your PCP before going somewhere else? | 3.92 (0.35) | 0.7/ 92.2 | 6 (0.8) | 2 (0.3) |
| **First contact – Access** | | | | |
| D1. When your PCP is open and you get sick, would someone from the practice see you on the same day? | 3.28 (0.7) | 1.7/ 39.3 | 2 (0.3) | 18 (2.5) |
| D2. When your PCP is open, can you get advice quickly over the phone if you need it? | 3.2 (0.81) | 3.9/ 35.2 | 8 (1.1) | 85 (11.9) |
| D3. When your PCP is closed, is there a phone number you can call when you get sick? | 3.26 (0.95) | 6.2/ 41.7 | 11 (1.5) | 149 (20.8) |
| D4. When your PCP is closed and you get sick during the night, would someone from the practice see you that night? | 1.67 (0.77) | 34/ 2.4 | 7 (1) | 201 (28.1) |
| **Ongoing Care** | | | | |
| E1. When you go to your PCP, are you taken care of by the same doctor or nurse each time? | 3.5 (0.69) | 1.4/ 59.2 | 5 (0.7) | 2 (0.3) |
| E2. If you have a question, can you call and talk to the doctor or nurse who knows you best? | 3.25 (0.81) | 2.9/ 40.6 | 9 (1.3) | 58 (8.1) |
| E3. Does your PCP know you very well as a person, rather than as someone with a medical problem? | 3.29 (0.86) | 3.5/ 51.3 | 9 (1.3) | 8 (1.1) |
| E4. Does your PCP know what problems are most important to you? | 3.55 (0.67) | 1.3/ 62.7 | 8 (1.1) | 10 (1.4) |
| **Coordination** | | | | |
| F1. Did your PCP discuss different places you could have gone to get help with that problem? | 3.41 (0.96) | 7.1/ 57.6 | 73 (10.2) | 26 (3.6) |
| F2. Did your PCP (or someone working with your PCP) help you make the appointment? | 2.82 (1.28) | 22.4/ 42 | 73 (10.2) | 21 (2.9) |
| F3. Did your PCP write down any information for the specialist about the reason for your visit? | 3.77 (0.64) | 2.7/ 74.3 | 74 (10.3) | 22 (3.1) |
| F4. After you went to the specialist or special service, did your PCP talk with you about what happened at the visit? | 3.63 (0.75) | 3.2/ 65.5 | 74 (10.3) | 19 (2.7) |
| **Comprehensiveness** | | | | |
| G1. Did your PCP discuss advice about healthy foods and unhealthy foods, or getting enough sleep | 2.96 (1.02) | 10.9/ 37.9 | 8 (1.1) | 15 (2.1) |
| G2. Did your PCP discuss home safety, like getting and checking smoke detectors and storing medicines safely | 1.91 (0.97) | 39.4/ 9.5 | 10 (1.4) | 35 (4.9) |
| G3. Did your PCP discuss ways to handle family conflicts that may arise from time to time | 1.99 (1.04) | 38.3/ 11.3 | 14 (2) | 42 (5.9) |
| G4. Did your PCP discuss advice about appropriate exercise for you | 3.01 (1) | 10.9/ 37.5 | 7 (1) | 17 (2.4) |
| G5. Did your PCP check on, and discuss the medications you are taking | 3.65 (0.69) | 2.9/ 74 | 2 (0.3) | 5 (0.7) |
| **Family-centeredness** | | | | |
| H1. Does your PCP ask you about your ideas and opinions when planning treatment and care for you or a family member? | 3.23 (0.94) | 6.9/ 49.2 | 7 (1) | 19 (2.7) |
| H2. Has your PCP asked about illnesses or problems that might run in your family? | 3.47 (0.82) | 4.5/ 60.4 | 5 (0.7) | 28 (3.9) |
| H3. Would your PCP meet with members of your family if you thought it would be helpful? | 3.48 (0.68) | 1.5/ 53.1 | 12 (1.7) | 36 (5) |

*(Continued)*

**Table 2.** (Continued)

| Subscale and item | Mean (SD) | Floor/ Ceiling effect | Frequency (%) | |
|---|---|---|---|---|
| | | | True missing (%) | Don't know/ can't remember (%) |
| **Community Orientation** | | | | |
| I1. Does anyone at your PCP's office ever make home visits? | 2.28 (1.11) | 20.8/ 12.6 | 6 (0.8) | 249 (34.8) |
| I2. Does your PCP know about the important health problems of your neighbourhood? | 2.73 (0.86) | 5.6/ 11 | 9 (1.3) | 270 (37.8) |
| I3. Does your PCP get opinions and ideas from people that will help to provide better health care? | 3.28 (0.71) | 2.4/ 29.9 | 9 (1.3) | 168 (23.5) |
| **Culturally Competent** | | | | |
| J1. Would you recommend your PCP to a friend or relative? | 3.83 (0.42) | 0.1/ 84.8 | 3 (0.4) | 1 (0.1) |
| J2. Would you recommend your PCP to someone who does not speak English well? | 3.38 (0.72) | 1.4/ 45.2 | 8 (1.1) | 70 (9.8) |
| J3. Would you recommend your PCP to someone who uses alternative medicine, such as herbs or homemade medicines, or has special beliefs about health care? | 3 (0.93) | 6.4/ 31.9 | 8 (1.1) | 76 (10.6) |

## Factor structure

Factor analysis was conducted on all 28 items. Three samples were used for factor analysis: (i) a sample of 180 participants with no imputation, (ii) a sample of 373 participants with developer-recommended imputation, and (iii) 606 participants with neutral value imputation.

(i)   Sample with no imputation

Bartlett's test of sphericity was significant ($\chi2$ (378) = 1875.635, $p < 0.001$), indicating that it was appropriate to use the factor analytic model on this set of data. The Kaiser-Meyer-Olkin measure of sampling adequacy indicated that the strength of the relationships among variables was high (KMO = 0.77); thus, it was acceptable to proceed with the analysis. The determinant of the correlation matrix was close to 0.0001 (S2 Table), suggesting potential multicollinearity in the non-imputed sample.

The CFA model did not fit the data well, indicated by a higher than acceptable SRMR ($\chi2$ (322) 487.551, $p < .001$, CFI 0.92, RMSEA 0.05, SRMR 0.10). All items exhibited standardised loadings > 0.50, with the exception of E1 (loading = 0.49), see S6 Table. In addition, the variance-covariance matrix of the estimated parameters was not positive definite, as the smallest eigenvalue was found to be negative (eigen = $-1.16e - 16$). We interpreted this eigenvalue to be effectively zero, and likely negative due to computational limitations. A zero eigenvalue may indicate model misspecification, or be an indication of linear dependency (redundancy). Furthermore, item J1 had a standardized loading of greater than 1.0 (loading = 1.19) along with a negative residual variance estimate ($-0.42$), indicating an improper solution, or "Heywood case". This suggests that the Culturally Competent latent variable explained more than 100% of the variance in item J1, which is not logically possible.

Since the data did not fit the hypothesized model, exploratory factor analysis was performed to further examine the factor structure of the PCAT-S. A scree plot of eigenvalues suggested eight factors should be retained. Varimax rotated factor loadings for an eight-factor solution are shown in S7 Table. The following deviations from the expected factor structure were observed: (i) item I1 loaded inadequately (loadings < 0.40) and items G1 and J1 demonstrated cross-loading, (ii) item E2 from the Ongoing Care subscale loaded onto a factor together with all four items from the First-Contact Accessibility subscale, and (iii) item G5 from the Comprehensiveness subscale loaded onto a factor together with the remaining items (E1, E3, and E4) from the Ongoing Care subscale.

**(ii) Sample with developer-recommended imputation**

The Kaiser-Meyer-Olkin measure of sampling adequacy was 0.84, and Bartlett's test of sphericity was significant ($\chi$2 (378) = 2721.387, $p < 0.001$), indicating the sample was appropriate for factor analysis. The CFA model demonstrated poor fit, with lower than acceptable CFI ($\chi$2 (322) 701.816, $p < .001$, CFI 0.89, RMSEA 0.06, SRMR 0.08). All items exhibited standardised loadings > 0.50, with the exception of D4, E1 and F2 (range 0.41–0.45, see S6 Table). Similar to the sample with no imputation, item J1 had a standardized loading of greater than 1.0 (loading = 1.24) along with a negative residual variance estimate (−0.54).

Results of exploratory factor analysis using varimax rotation with an eight factor solution are shown in S8 Table. Items loaded onto factors as expected with the following exceptions: (i) item F4 and I1 loaded inadequately (loadings < 0.40), (ii) item E2 from the Ongoing Care subscale loaded onto a factor together with all four items from the First-Contact Accessibility subscale, and (iii) item G5 from the Comprehensiveness subscale, and item J1 from the Culturally Competent subscale, loaded onto a factor together with the remaining items (E1, E3, and E4) from the Ongoing Care subscale.

**(iii) Sample with neutral-value imputation**

The Kaiser-Meyer-Olkin measure of sampling adequacy was 0.85, and Bartlett's test of sphericity was significant ($\chi$2 (378) = 4171.491, $p < 0.001$), indicating the sample was appropriate for factor analysis. The confirmatory factor analysis model demonstrated acceptable fit ($\chi$2 (322) 972.528, $p < .001$, CFI = 0.90, RMSEA 0.06, SRMR 0.07). All items exhibited standardised loadings > 0.50, with the exception of D3, D4, E1 and F2 (range 0.36–0.50, see S6 Table). Similar to the sample with no imputation, the smallest eigenvalue was effectively zero (eigen = 1.5 $e$ − 17) and item J1 had a standardized loading of greater than 1.0 (loading = 1.13) along with a negative residual variance estimate (−0.28).

CFA fit indices fell within an acceptable range; however, two of the fit indices indicated only borderline fit, and we therefore still conducted an EFA to further investigate factor structure. We conducted an EFA as an additional exploratory step to assess whether the data might indicate a more appropriate factor structure, and to provide a comparison across the three imputation samples. Results of exploratory factor analysis using varimax rotation with an eight factor solution are shown in S9 Table. Items loaded onto factors as expected with the following exceptions: (i) items D3, E1, F2, I1 and all three items from the Family Centeredness subscale (H1, H2, and H3) loaded inadequately (loadings < 0.40), (ii) item E2 from the Ongoing Care subscale loaded onto a factor together with the remaining three items from the First-Contact Accessibility subscale (D1, D2 and D3), (iii) item F4 from the Coordination subscale loaded onto a factor together with three items from the Comprehensiveness subscale (G1, G4 and G5), while items G2 and G3 loaded onto a separate factor altogether, and (iv) item J1 from the Culturally Competent subscale, loaded onto a factor together with the remaining items (E1, E3, and E4) from the Ongoing Care subscale.

## Internal consistency

Cronbach's alpha showed acceptable internal consistency for First-Contact Utilization (2 items, $\alpha = 0.71$), Ongoing Care (4 items, $\alpha = 0.72$), and Comprehensiveness (5 items, $\alpha = 0.77$). Internal consistency for the remaining subscales were weak (range 0.61–0.67). Results are presented in Table 3.

## Discussion

This study examined the reliability and validity of the PCAT-S as a measure of patient experiences within general practices in Australia. Our findings were mixed, with only three subscales demonstrating good internal consistency: Comprehensiveness, Ongoing Care and First Contact – Utilization. The remaining subscales exhibited weak internal consistency. Examination of factor structure using CFA identified possible model misspecification. Subsequent exploratory factor analysis through EFA indicated that, generally, the items supported the hypothesized factor structure, although some deviations

**Table 3. Internal consistency.**

| Subscale | Cronbach's alpha |
|---|---|
| First contact – Utilization | 0.71 |
| First contact – Access | 0.61 |
| Ongoing Care | 0.72 |
| Coordination | 0.61 |
| Comprehensiveness (services provided) | 0.77 |
| Family-centeredness | 0.64 |
| Community Orientation | 0.65 |
| Culturally Competent | 0.61 |

Cronbach's alpha measures internal consistency. Scores >0.7 indicate acceptable reliability. Results were obtained using complete observations only.

were observed. For each subscale, we discuss the implications of our findings and propose potential explanations. Readers should note that we did not conduct formal face validity assessments. However, we attempted to relate our findings to contextual factors, which we believe have provided useful insights for interpreting the results and potential areas for future research.

In the **First Contact – Utilisation subscale** (items C1–2), both items exhibited a large ceiling effect, with greater than 90% of participants selecting the highest response for both items. This may be because most participants had attended their GP or practice for over 5 years, hence would naturally be more inclined to choose their regular practice for checkups or new health issues. Items with large ceiling effects can compromise statistical assumptions and suggest problems with the instrument (limited range) or response bias (unbalanced sample). Despite this limitation, the subscale showed adequate factor loading in both CFA and EFA, and acceptable internal consistency.

For the **First Contact – Access subscale** (items D1–4), participants scored generally positively for the first three items (mean range 3.2–3.28) and negatively for the last item (mean 1.67). Interestingly, for this last item '*When your PCP is closed and you get sick during the night, would someone from the practice see you that night?*', almost 30% of participants chose 'Don't know/ don't remember'. This may be due to the fact that participants would only know if they had experienced such an event. For items D3 and D4, factor loadings varied across EFA and CFA, and across samples, with some factor solutions showing lower than acceptable loadings. Coupled with weak internal consistency ($\alpha = 0.61$) for this subscale, these results suggest items, particular D3 and D4, may need revision. These two items specifically address accessibility to after-hours care, which in Australia is often not provided by a patient's regular GP but by services such as extended hours GP services, medical deputizing services (MDS), nurse-led walk-in clinics, or public emergency departments [42]. This suggests a need to reconsider these items to better capture the reality of after-hours care accessibility in the Australian healthcare context which now includes urgent care clinics [43].

Across all three imputation method samples, EFA of the **Ongoing Care subscale** (E1–4), consistently showed item E2 '*If you have a question, can you call and talk to the doctor or nurse who knows you best?*' loaded onto the First Contact – Access subscale. This was not entirely unexpected as this question also relates to accessibility, as it implies a need to make a phone call from home. Construct overlap between First-Contact Accessibility and Ongoing Care has also been observed in other PCAT studies [23,26,44,45], with some researchers collapsing these into a single subscale [23,46]. Despite these issues with the factor analysis, the Ongoing Care subscale showed acceptable internal consistency ($\alpha = 0.72$), suggesting items generally measure the same construct.

For the **Coordination** subscale (items F1–4), CFA and EFA generally found items F2 '*Did your PCP (or someone working with your PCP) help you make the appointment?*' and F4 '*After you went to the specialist or special service, did*

*your PCP talk with you about what happened at the visit?* loaded inadequately. The subscale exhibited weak internal consistency (α = 0.61), indicating potential issues with items aligning to a single construct. These findings suggest a need to further examine the wording or relevance of items F2 and F4 in particular.

The **Comprehensiveness** subscale performed relatively well, showing the highest internal consistency of all subscales (α = 0.77) and adequate factor loadings across all three CFA. However, EFA found deviations from the hypothesized data structure. In the no imputation and developer-recommended imputation samples, item G5 '*Checking on, and discussing the medications you are taking*' loaded on the Ongoing Care subscale. G5 differs from other items in this subscale, as it relates to clinical quality (tasks that should be done) rather than comprehensive care, which encompasses a broader range of services (tasks that could be done). Medication management likely becomes more relevant as continuity and the patient-provider relationship improve [47]. In the neutral-value imputation sample, items G2 (home safety) and G3 (family conflicts) loaded onto another factor entirely. In the Australian context, discussions around safety and family conflict may not be common in general practice settings. In different country versions of the PCAT, items in the Comprehensiveness subscale differ widely, and are selected based on population demographics (for example, in our study which focuses on higher-risk patients, items from the original scale covering childhood immunisations were excluded) [40]. In light of this, Haggerty et al (2011) [48] argued that providers, rather than patients, are best positioned to assess this particular subscale, primarily because providers plan care for a broad range of patients, while patients can only assess services based on their personal experience.

**Family-centeredness** (items H1–3), showed weak internal consistency and mixed results for factor analysis, with neither item loading adequately in EFA of the neutral-value sample, and adequate loading in the other non-imputation and developer-recommended samples. Based on these findings it is evident that the imputation approach used influenced the factor solution, making subsequent conclusions about the validity of this subscale uncertain. Analytical issues arising from lack of clarity on how to score 'Don't know/don't remember' responses are a known problem with the PCAT. Researchers have used various methods, including neutral value [8,22], median value [38], or a mix of approaches [23,24,26]. This issue has led to calls for further work and international collaborations to refine response options to better reflect patient experiences in different contexts [23,26].

All **Community Orientation** items had high 'Don't know/don't remember' responses, suggesting that patients may not be the best information source for this subscale. Over one third of participants responded 'Don't know/don't remember' for item I1 '*Does anyone at your PCP's office ever make home visits?*' and item I2 '*Does your PCP know about the important health problems of your neighbourhood?*'. This could be because participants never found the need to inquire about whether their GP conducts home visits or is aware of community issues. Notably, in Australia, home visits have been on the decline [49,50]. Additionally, Australian GPs primarily focus on providing individual healthcare, while community healthcare is predominantly managed by public health agencies and community health centres [51]. Furthermore, in the EFA, item I1 showed near acceptable loadings (range 0.30–0.40) in both Factor 2 (First-Contact Accessibility) and Factor 7 (Community Orientation) across all three samples. This finding aligns with previous findings by Haggerty et al (2011) [26], which found a construct overlap between First-Contact Accessibility and Community Orientation.

CFA and EFA were consistent across the three imputation methods for the **Culturally Competent** (items J1–3) subscale. As mentioned earlier, in the original PCAT validation study Culturally Competent emerged as a distinct factor only in a predominantly non-white sample [7]. Items J1–J3 may be more challenging to interpret in populations with less diversity. In our study, we found item J1 '*Would you recommend your PCP to a friend or relative?*' showed a high ceiling effect, with 85% of participants selecting the highest option. Again, this may be because of this sample of participants already having a long-term relationship with their GP (over 75% of participants had a relationship of 5 years or more) and selection bias (participants agreeing to participate in the study may have been more inclined to have positive experiences with the practice). Hence, participants would be more likely to recommend their regular practice to others. All EFAs consistently

found item J1 as loading onto factor 3 (Ongoing Care). This is expected, as both liking and recommending a practice are indicative of good relational continuity.

Limitations relating to the sample of participants have been discussed previously [13,14]. Specific limitations regarding this analysis of the PCAT-S include the following: the results may not be generalizable to the wider Australian population due to the specific sample used. The sample consisted of individuals who (i) generally had a long-term relationship with their GP or practice, and (ii) were adult higher-risk patients due to chronic disease and/or age. Notably, our sample were predominantly older adults with a mean age of over 60 years. These factors may have led to higher scores in areas such as Ongoing Care, as participants likely had more established relationships with their healthcare providers. Older patients or patients with chronic disease also tend to have more complex or frequent interactions with healthcare services, which may result in different experiences with primary care compared to younger, healthier individuals who have fewer or less involved healthcare engagements. In addition, nearly all items exhibited floor or ceiling effects, likely reflecting the homogeneity of our sample.

For factor analysis, we excluded subjects with 'Don't know/can't remember' responses, which reduced statistical power and may have produced a biased sample. To address this, we conducted analyses with imputed samples, which altered our overall conclusions for some factors. Despite these limitations, this study provides a solid foundation for further work on the PCAT-S in Australia. Future research could explore the correlation of PCAT-S subscales with other validated Australian instruments that measure theoretically related constructs, such as the PAIS [11] or PESS [12].

Our CFA identified notable limitations, including zero eigenvalues and Heywood cases, which undermine the credibility of the results. A zero eigenvalue may indicate model misspecification, or be an indication of linear dependency or redundancy, and Heywood cases imply that unique factors have negative error variances. Negative error variances often indicate model misspecification, but there can be other reasons, such as random sampling fluctuations, small sample size, too few items for a given factor, or too many common factors to provide stable estimates with the available data. In Berra et al. (2011) [45], Heywood cases were observed in a factor solution examining the original factor structure, which included six original subscales. As is typical when model misspecification is suspected, the authors further examined several factor solutions and found a reduced five-factor model showed no Heywood cases and adequate fit. Exploring factor solutions with varying numbers of factors was beyond the scope of this study and would be more appropriate for face and content validity assessments, which involve item selection and refinement. We also acknowledge that this study does not assess convergent and discriminant validity, which would have been helpful for clarifying factor analyses. Instead, we further examined factor structure via EFA, which provides insights into the underlying factor structure without the constraints of a predefined model – this can be useful for understanding the causes of model misspecification observed in CFA and provides a more comprehensive understanding of the factor structure.

One subscale (*First-Contact Utilisation*) included only two items – while this subscale contained three items in the original PCAT-S, one item was omitted for the purpose of the larger RCT from which our study data is drawn from. We acknowledge that while we retained it to reflect how the instrument was used in practice, this may have contributed to poorer model fit and may have affected the overall CFA model.

We note that in studies which performed CFA for psychometric validation of country-specific adaptations of the PCAT, best fitting models demonstrated goodness-of-fit statistics well within acceptable ranges [23,44,45]. Such adaptations of the PCAT involve modifications to survey questions, rearranging of items into different subscales or the addition or removal of items [8,24,38,40,45,52]. In contrast, the version used in our study was pragmatically modified for use in a larger RCT, and these changes were not informed by formal psychometric procedures such as content validation. Our findings that certain subscales require revision to improve reliability and validity, suggest a need to develop an Australian version of the PCAT-S, which is consistent with international evidence emphasising the importance of adapting the PCAT to local healthcare contexts.

 

Results from our study can be compared to Canadian research on the PCAT-S [26,53]. Like our study, the instrument was evaluated without prior refinement; additionally, the study was conducted in Canada, which shares a similar healthcare context to Australia compared to other validation studies conducted in Asia, Africa and South America [22–25,38,40,44,45,52,54–56]. Internal consistency scores were similar to our study for several subscales: First-Contact – Utilization (0.68 [Canadian study] vs. 0.71 [our study]), Ongoing Care (0.73 vs. 0.72), Comprehensiveness (0.72 vs. 0.77) and Community Orientation (0.65 vs. 0.65) [26], however, the Canadian study reported higher scores for First contact – Access (0.72 vs. 0.61) and Coordination (0.76 vs. 0.61) [26]. These differences may reflect variations in healthcare delivery between Australian general practices and Canada – for example, out-of-hours access may be less common in Australia.

At present, there are no validated instruments specifically designed for evaluating primary care patient experiences in Australia (except for PAIS and PESS, which focus on different domains [57]), reinforcing the PCAT's value as a tool for this purpose. Our initial validation of the PCAT-S in an Australian general practice sample showed mixed psychometric performance. While some subscales demonstrated acceptable reliability and model fit, others fell below recommended reliability thresholds. Hence survey results should be interpreted with nuance, and consideration of item relevance and interpretation in the Australian context.

Next steps should include the development of a country-specific Australian version of the PCAT. As in other countries, this process should be guided by formal face and content validity assessments, expert consultation, and engagement with patients and providers. Item refinement should include reviewing relevance of items which may not be relevant to how general practice is currently delivered in Australia (such as after-hours care, home visits, general practice involvement in community health services, discussion of home safety, and facility-related conflict). Consideration should also be given to how provider and specialist services operate within the Australian healthcare system. Future development work should draw on data collected specifically for psychometric purposes and aim for broader demographic representation. More advanced psychometric testing, including assessments of convergent and discriminant validity and exploration of alternative factor structures, would further strengthen the instrument.

A version of the PCAT-S validated for the Australian context would provide a robust tool for assessing primary care performance and facilitate meaningful national and international comparisons.

## Supporting information

**S1 Table. Adjustments to the PCAT-S for the EQuIP-GP trial.**
(DOCX)

**S2 Table. Multicollinearity statistics.** The determinant of the correlation matrix that is smaller than 0.00001 suggests an issue with multicollinearity.
(DOCX)

**S3 Table. Factor correlations of extracted factors from Exploratory Factor Analysis (EFA) using oblimin rotation, complete observations only (n = 180).** Correlations where $r > |0.30|$) are indicated by an asterisk.
(DOCX)

**S4 Table. Factor correlations of extracted factors from Exploratory Factor Analysis (EFA) using oblimin rotation, developer-recommended imputation (n = 373).** Correlations where $r > |0.30|$) are indicated by an asterisk.
(DOCX)

**S5 Table. Factor correlations of extracted factors from Exploratory Factor Analysis (EFA) using oblimin rotation, neutral-value imputation (n = 606).** Correlations where $r > |0.30|$) are indicated by an asterisk.
(DOCX)

**S6 Table. Standardised factor loadings from CFA models by imputation method.** Confirmatory Factor Analysis (CFA). Problematic items are indicated by an asterisk. Items were considered to load onto a specific factor if the

standardised factor loading was > 0.50. Heywood cases occur when the loading is greater than 1.0. Items are presented using wording from the administered survey.
(DOCX)

**S7 Table. Factor loadings and extracted factors using complete observations only (n = 180).** Exploratory factor analysis (EFA) using varimax rotation. The highest loadings for each item are bolded. Items were considered to load onto a specific factor if the factor loading was > 0.40 for that factor and <0.40 for all other factors. Items are presented using wording from the administered survey.
(DOCX)

**S8 Table. Factor loadings and extracted factors, developer-recommended imputation (n = 373).** Exploratory factor analysis (EFA) using varimax rotation. The highest loadings for each item are bolded. Items were considered to load onto a specific factor if the factor loading was > 0.40 for that factor and <0.40 for all other factors. Items are presented using wording from the administered survey.
(DOCX)

**S9 Table. Factor loadings and extracted factors using neutral-value imputation (n = 606)** . Exploratory factor analysis (EFA) using varimax rotation. The highest loadings for each item are bolded. Items were considered to load onto a specific factor if the factor loading was > 0.40 for that factor and <0.40 for all other factors. Items are presented using wording from the administered survey.
(DOCX)

## Acknowledgments

We would like to thank the Royal Australian College of General Practitioners (RACGP), the Australian Government Department of Health, as well as the participating general practices and patients involved in the larger EQuIP-GP project which provided the data for this study.

## Author contributions

**Conceptualization:** Marijka J. Batterham, Andrew Bonney.

**Data curation:** Marijka J. Batterham, Andrew Bonney.

**Formal analysis:** Chau M. Bui, Marijka J. Batterham, Andrew Bonney.

**Funding acquisition:** Judy Mullan, Gregory Peterson, Jan Radford, Simon Eckermann, Danielle Mazza, Grant Russell, Andrew Bonney.

**Investigation:** Judy Mullan, Gregory Peterson, Christine Metusela, Jan Radford, Danielle Mazza, Grant Russell, Andrew Bonney.

**Methodology:** Chau M. Bui, Marijka J. Batterham, Simon Eckermann, Andrew Bonney.

**Project administration:** Judy Mullan, Gregory Peterson, Christine Metusela, Jan Radford, Danielle Mazza, Grant Russell, Andrew Bonney.

**Supervision:** Marijka J. Batterham, Judy Mullan, Gregory Peterson, Simon Eckermann.

**Validation:** Marijka J. Batterham, Andrew Bonney.

**Writing – original draft:** Chau M. Bui, Danielle Mazza.

**Writing – review & editing:** Chau M. Bui, Marijka J. Batterham, Judy Mullan, Gregory Peterson, Christine Metusela, Jan Radford, Simon Eckermann, Grant Russell, Andrew Bonney.

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
