## [Decision Letter · Decision Letter 0]

21 May 2025

Thank you for submitting your manuscript to PLOS ONE. After careful consideration, we feel that it has merit but does not fully meet PLOS ONE’s publication criteria as it currently stands. Therefore, we invite you to submit a revised version of the manuscript that addresses the points raised during the review process.

We look forward to receiving your revised manuscript.

Kind regards,

Bojana Bukurov, M.D., Ph.D.

Academic Editor

PLOS ONE

Journal Requirements:

2. Please note that funding information should not appear in the Acknowledgments section or other areas of your manuscript. We will only publish funding information present in the Funding Statement section of the online submission form. Please remove any funding-related text from the manuscript.

“I have read the journal's policy and the authors of this manuscript have the following competing interests: Prof Grant Russell, A/Prof Jan Radford, and Prof Danielle Mazza have received honoraria from the RACGP for expert committee roles. The other authors have no conflicts to declare.”

5. In this instance it seems there may be acceptable restrictions in place that prevent the public sharing of your minimal data. However, in line with our goal of ensuring long-term data availability to all interested researchers, PLOS’ Data Policy states that authors cannot be the sole named individuals responsible for ensuring data access (http://journals.plos.org/plosone/s/data-availability#loc-acceptable-data-sharing-methods).

Reviewers' comments:

Reviewer's Responses to Questions

**Comments to the Author**

1. Is the manuscript technically sound, and do the data support the conclusions?

Reviewer #1: Partly

Reviewer #2: Partly

2. Has the statistical analysis been performed appropriately and rigorously?

Reviewer #1: No

Reviewer #2: Yes

3. Have the authors made all data underlying the findings in their manuscript fully available?

Reviewer #1: No

Reviewer #2: No

4. Is the manuscript presented in an intelligible fashion and written in standard English?

Reviewer #1: Yes

Reviewer #2: Yes

Reviewer #1: This paper describes the psychometric properties of an instrument that measures patients experience with primary care. While the paper was interesting to read, I do have some questions and suggestions for improvements.

In the introduction several psychometric properties from other studies are reported following the aim of the study. The last sentence gives a rationale for the study. I would recommend moving these reports from other studies to the method section and allowing the introduction to end in an aim.

In the introduction I would rather see more information on the PCAT in general, when, how, and why was it developed for instance. It is described that it is one of the most widely used tools, that sounds great, but again, as a reader I need more information to be able to understand why and how this tool is useful.

From this paper it is unclear where I can find information on the development and validation of the original PCAT-scale, and its short form. This should be clearly stated with a reference when the scale is first introduced, and again in the method section, making it possible for us reviewers and later also readers to read the original study, in order to understand the theory behind the instrument, how items have been selected into the final version, if it has been analyzed for convergent and discriminant validity and so on.

Data analysis, when describing the test-retest reliability the retesting interval is much wider than the usually recommended 2 weeks. If this method is intended to indicate temporal stability, this needs to be more clearly stated. Why should assessments one year apart show such stability? Either remove these analyses or give justification, if possible. In the discussion it gets even more confusing, when authors try to describe why temporal stability was low. The idea of test-retest is that the latent variable should be stable over the time frame for reassessment, to enable conclusions on variability or stability of results.

I think the paper would benefit from adding an analysis of item-scale correlations, to discern both convergent and discriminant validity. Perhaps this would make the inconclusive results from the factor analyses clearer and add important information on the subscales.

In the result section, page 24, this CFA did show acceptable model fit indices?

How does the results from these CFA compare to other studies? One subscale has inadequate number of items for CFA, how have other studies on this scale described this limitation? How can this have impacted the results? Should the analyses be adjusted in some way?

Moreover, when reading the reference to the Vietnamese version, the letters describing each subscale differs, how come? For instance, culturally competent has the letter “J” in this paper, but “K” in the Vietnamese version, making comparisons confusing.

I found the conclusion of this paper confusing. As the title suggest, this paper will describe the psychometric properties of the PCAT-S in a specified sample. Then the conclusion is that findings were mixed, and more research is needed, however the scale could be used in its current version in Australia. With reliability levels below recommended values, how would that impact interpretations of results?

To the best of authors knowledge from reading validation studies from other countries, combined with the results from the current study, what are the next steps? From these results, what adaptations do you suggest, and how should these be addressed in future research?

Reviewer #2: Summary of research

The manuscript reports a psychometric analysis of the Primary Care Assessment Tool, Short Form (PACT-S), an existing instrument measuring patients’ experience with their primary care provider (PCP), in the general practice setting in Australia. The purpose of the study was to examine the factor structure and reliability of the PACT-S.

The sample of 715 adults comprised a subgroup of participants in a larger RCT and was drawn from 34 practices across Australia. The sample included adults who were aged 64 or younger with chronic illnesses or who were age 65 or above. Analyses consisted of confirmatory factor analyses (CFA) to determine how well the data fit the factor structure proposed by the instrument’s authors and subsequent exploratory factor analyses (EFA).

Analyses also included internal consistency reliability (Cronbach’s alpha) and test-retest reliability (Intraclass Correlation Coefficient; ICC). Two different models for imputation were used, in addition to a model with no imputation of missing values. Helpfully, the authors include item wording and response options for the full instrument and clearly label items from each subscale.

The authors conducted all analyses correctly and transparently. Relevant decision steps in conducting the analyses were reported clearly, as were the results. Results are broken down by subscale in the discussion section, and possible interpretations and future steps provided for each.

Recommendation

The manuscript provides clear relevance to practice and clearly reports the data collection, data analysis, results, and interpretation. All analyses conducted appear reasonable and correct. The authors provide justification for each analytical decision. The work represents an important contribution to instrument validation which, as the authors point out, is essential for conducting research on health systems and health policies in cross-national contexts. The discussion is helpful for putting the results in context. The authors’ suggested interpretations and directions for future research are reasonable. I recommend publication but I hope the authors will consider my comments as I believe the manuscript can be strengthened if these are addressed.

Minor concerns

I appreciate the authors’ providing a rationale for using the neutral-value imputation scheme, bolstered by citing multiple prior publications that used this method (lines 188-191). Presenting three sets of results adds complexity that may not be of interest to all readers. Perhaps some information could be transferred to supplemental tables.

The rationale for conducting EFA could be strengthened. The authors have already established the factor structure varies across different national health care contexts, based on the citations provided and the discussion on lines 94-104. If we already know the factor structure is not invariant across nations, do we need further evidence of this? This rationale could be clarified.

Would it be possible to report what proportion of patients met inclusion criteria by virtue of their age alone (65 years or older) versus being in the 18-64 age group and having a chronic illness? This might be helpful when interpreting the mean (SD) age in Table 1; 66.9 years seems somewhat low and could indicate a substantial portion of the sample were younger than 65.

It would be useful to more closely examine the data to identify causes for poor model fit (especially in the CFA). The sample size is inadequate for the model with no imputation but adequate for both imputation models, so isn’t a likely culprit. However, multicollinearity statistics would be helpful for the reader (and could be included in the supplemental materials). Reducing the number of factors for the EFA could also improve model fit.

The list of exceptions seems rather lengthy when stating the factor structure of the EFA (e.g., lines 263-267). Beginning by pointing out five items (of 28) did not load as expected would frame things differently. The authors’ existing language is correct, but it caused me to mark the margin with, “that’s an awful lot of exceptions!”

Including only participants in the control arm in the test-retest analysis is entirely appropriate. Is it possible to state the mean (median, range) length of time elapsed between the two measurement points? This may have been reported elsewhere but it’s key to interpreting the test-retest results.

The authors state reducing the number of factors is beyond the scope of this study—fair enough. However, the rationale was that changing the number of factors would be more appropriate for “face and content validity assessments” (line 468). Contrast this with the Discussion section, which seems to discuss the face validity of numerous individual items. (The term ‘face validity’ is not used, but that seems to be what’s happening.) Some of this discussion seems to go beyond the data. Rewording or reframing could resolve this apparent discrepancy.

The authors reference “poor temporal stability (α = 0.32).” This value is the ICC reported in Table 7. Would it be clearer to use rho in place of alpha, here?

Is it possible to rule out having changed one’s GP as an explanation for the low test-retest reliability of the “First contact – Utilization” subscale? The authors posit changing providers as a potential explanation (lines 357-361). It’s possible the larger RCT has data on whether a participant switched practices. Did participants in the control arm have a shorter duration of relationship with the GP (or practice) than those in the treatment arm at baseline? Given that more than three-quarters of participants had been with their GP (or practice) over 5 years, changing providers doesn’t seem an especially compelling explanation for the low ICC on this subscale.

**Do you want your identity to be public for this peer review?** For information about this choice, including consent withdrawal, please see our Privacy Policy

Reviewer #1: No

Reviewer #2: **Yes:** Susan L. Schoppelrey

---

## [Author Response · Author response to Decision Letter 1]

4 Aug 2025

Please see "Response to Reviewers" file for a formatted version of this response.

Part II: Response to Reviewer#1 comments

Reviewer #1: This paper describes the psychometric properties of an instrument that measures patients experience with primary care. While the paper was interesting to read, I do have some questions and suggestions for improvements. In the introduction several psychometric properties from other studies are reported following the aim of the study. The last sentence gives a rationale for the study. I would recommend moving these reports from other studies to the method section and allowing the introduction to end in an aim.

Authors’ response: Thank you for the suggestion. We agree that this is a useful change to our manuscript and have now moved reports from other studies to the Method section, and the Introduction now ends in an aim.

Reviewer #1: In the introduction I would rather see more information on the PCAT in general, when, how, and why was it developed for instance. It is described that it is one of the most widely used tools, that sounds great, but again, as a reader I need more information to be able to understand why and how this tool is useful.

From this paper it is unclear where I can find information on the development and validation of the original PCAT-scale, and its short form. This should be clearly stated with a reference when the scale is first introduced, and again in the method section, making it possible for us reviewers and later also readers to read the original study, in order to understand the theory behind the instrument, how items have been selected into the final version, if it has been analyzed for convergent and discriminant validity and so on.

Authors’ response: More information on the PCAT in general was added to the Introduction, including some statements about why the tool is useful. We added additional references to the first research papers around the development and validation of the original PCAT-scale and its short form.

Lines 52 to 71: “The Primary Care Assessment Tool (PCAT) is widely used for evaluating primary care services globally, was developed by the Johns Hopkins Primary Care Policy Centre [4]. The PCAT has versions that are intended for consumers, including children, and health care providers. and is designed to assess the extent and quality of primary care services in provider settings identified by consumers as their predominant source of health care.

The PCAT is based on a theoretical model of primary care formulated by Starfield [5], which incorporates the following four essential primary care attributes: first-contact accessibility and use, continuity, comprehensiveness, and coordination, as well as family-centeredness, community orientation, and cultural competence.

The development of the PCAT began with the Children and Adolescents version, which was initially validated in the United States by Cassady et al. (2000) [6]. This was followed by the development and validation of the Adult Edition and the Adult Short Form, with further validation and refinement documented by Shi et al. (2001) [7].

The PCAT has since been translated, validated, and utilized across many countries, including Canada, Brazil, Malta, Spain, South Korea, Japan, China, Hong Kong, Taiwan, Tibet, Vietnam, Malawi, Uganda, and South Africa [4–8]. As noted by Rocha and colleagues [8], while many tools assess specific dimensions of primary care quality, there are few like the PCAT that enable a comprehensive evaluation of primary care from the population perspective. Additionally, the Consumer/Client Surveys are designed for self-administration and require only a high school reading level, making the tool broadly accessible to the general population [7]. ”

Reviewer #1: Data analysis, when describing the test-retest reliability the retesting interval is much wider than the usually recommended 2 weeks. If this method is intended to indicate temporal stability, this needs to be more clearly stated. Why should assessments one year apart show such stability? Either remove these analyses or give justification, if possible. In the discussion it gets even more confusing, when authors try to describe why temporal stability was low. The idea of test-retest is that the latent variable should be stable over the time frame for reassessment, to enable conclusions on variability or stability of results.

Authors’ response: Thank you. This is a valid query of our study design, as a one-year interval is relatively long for measuring test-retest reliability. We have provided some additional justification for using this interval:

Lines 223 to 229: “Although test-retest reliability is typically assessed over short intervals, typically one to two weeks, to minimize the influence of external changes, longer intervals can still provide valuable insights in the context of longitudinal research. This is relevant for the PCAT, which has been used in longitudinal studies, and in measuring national and state-level primary care experiences over time in Brazil and Canada [13, 42-45]. Instruments that consistently reproduce the same result across extended periods are considered to have high temporal stability, which can be useful for longitudinal research.”

Reviewer #1: I think the paper would benefit from adding an analysis of item-scale correlations, to discern both convergent and discriminant validity. Perhaps this would make the inconclusive results from the factor analyses clearer and add important information on the subscales.

Authors’ response: We thank the reviewer for the suggestion regarding item–scale correlations. While we agree that such an analysis could provide additional insights, we have opted not to include it in the current manuscript to maintain focus and clarity, given the already substantial length and number of analyses presented. We have, however, acknowledged this as a limitation in the manuscript and noted it as an area for future research.

Lines 484 to 486: “We also acknowledge that the study did not assess convergent and discriminant validity, which could have been helpful for clarifying the results of the factor analyses.”

Reviewer #1: In the result section, page 24, this CFA did show acceptable model fit indices?

Authors’ response: Yes, this is correct, CFA fit indices fell within acceptable range for the neutral-value imputation sample and EFA is not indicated in this situation. However, two of these fit indices were only borderline fit (CFI = 0.90, RMSEA 0.06, SRMR 0.07; compared to benchmarks CFI ≥0.90, SRMR <0.08, and RMSEA ≤0.06). The hypothesized model remains plausible but may benefit from further refinement. We conducted an EFA as an additional exploratory step to assess whether the data might indicate a more appropriate factor structure, and to provide a comparison across the three imputation samples.

Lines 319 to 320: “CFA fit indices fell within an acceptable range; however, two of the fit indices indicated only borderline fit, and we therefore still conducted an EFA to further investigate factor structure.”

Reviewer #1: How does the results from these CFA compare to other studies? One subscale has inadequate number of items for CFA, how have other studies on this scale described this limitation? How can this have impacted the results? Should the analyses be adjusted in some way?

Authors’ response: In studies which performed CFA for psychometric validation, best fitting models demonstrated goodness-of-fit statistics well within acceptable ranges – for example Berra et al 2011, Kijima et al 2021 and Wang et al 2014. However, these CFA models evaluated country-specific adaptations of the PCAT, rather than the PCAT in it’s original form. In our study, the version of the tool we examined was not an Australian adaption of the PCAT – rather, it was a version that was pragmatically modified for use in a larger RCT and these modifications were not guided by formal psychometric methods.

The subscale in question is First-Contact Utilization, which contains three items in the original PCAT Adult Short Form. One item was omitted for the purpose of the larger RCT (from which our study data is drawn from – see Table S1). However, we chose to retain all subscales in the presentation of our results, including First-Contact Utilisation, because our primary aim was to explore how an existing, previously validated instrument performs in a new context, rather than conduct a comprehensive validation of a new instrument. We recognise that the reduced item count may have contributed to the poorer model fit observed for this subscale and may have affected the overall CFA model. However, adjusting the model by removing the subscale would deviate from our intent to assess the tool as used in the RCT.

We have revised our Discussion accordingly:

Lines 491 to 506: “One subscale (First-Contact Utilisation) included only two items – while this subscale contained three items in the original PCAT-S, one item was omitted for the purpose of the larger RCT from which our study data is drawn from. We acknowledge that while we retained it to reflect how the instrument was used in practice, this may have contributed to poorer model fit and may have affected the overall CFA model.

We note that in studies which performed CFA for psychometric validation of country-specific adaptations of the PCAT, best fitting models demonstrated goodness-of-fit statistics well within acceptable ranges [24, 48, 49]. Country-specific adaptations of the PCAT involve modifications to survey questions, rearranging of items into different subscales or the addition or removal of items [8, 25, 37, 39, 49, 56]. In our study, the version of the tool we examined was not a country-specific adaption of the PCAT – rather, it was a version that was pragmatically modified for use in a larger RCT and these modifications were not guided by formal psychometric methods such as content validation assessments. Our findings—that certain subscales require revision to improve reliability and validity—suggest a need to develop an Australian version of the PCAT-S. This is consistent with the broader PCAT literature, which highlights the importance of country-specific adaptation to ensure alignment with local healthcare experiences.”

Reviewer #1: Moreover, when reading the reference to the Vietnamese version, the letters describing each subscale differs, how come? For instance, culturally competent has the letter “J” in this paper, but “K” in the Vietnamese version, making comparisons confusing.

Authors’ response: We agree with the reviewer that this discrepancy is confusing. The Vietnamese version is based on the PCAT Adult Expanded version, while our analysis used the PCAT Adult Short Form and additionally excludes two subscales: Coordination (Information Systems) and Comprehensiveness (Services Available) (see Supplementary Table S1). As a result, the alphabetical ordering of the subscales differs between the two versions. To clarify this for readers, we have added a note in the Methods section.

Lines 124 to 127: “Note the alphabetical ordering of the subscales from the original PCAT-S has not been retained due to the exclusion of two subscales—Coordination (Information Systems) and Comprehensiveness (Services Available) (see Supplementary Table S1).”

Reviewer #1: I found the conclusion of this paper confusing. As the title suggest, this paper will describe the psychometric properties of the PCAT-S in a specified sample. Then the conclusion is that findings were mixed, and more research is needed, however the scale could be used in its current version in Australia. With reliability levels below recommended values, how would that impact interpretations of results?

Authors’ response: Thank you for this helpful feedback. We agree that our conclusion required clarification and have revised this section of the manuscript (see below).

Lines 517 to 524: “At present, there are no validated instruments specifically designed for evaluating primary care patient experiences in Australia (except for PAIS and PESS, which focus on different domains [61]), reinforcing the PCAT's value as a tool for this purpose. This highlights the value of the PCAT as a potential tool for this purpose. Our initial validation of the PCAT-S in an Australian general practice sample showed mixed psychometric performance. While some subscales demonstrated acceptable reliability and model fit, others fell below recommended reliability thresholds. Hence survey results should be interpreted with nuance, and consideration of item relevance and interpretation in the Australian context.

Reviewer #1: To the best of authors knowledge from reading validation studies from other countries, combined with the results from the current study, what are the next steps? From these results, what adaptations do you suggest, and how should these be addressed in future research?

Authors’ response: We have added to the conclusion to outline the next steps for research and instrument development. This includes specific recommendations for item refinement, methodology, and psychometric testing.

Lines 525 to 539: “Next steps should include the development of a country-specific Australian version of the PCAT. As with other country-specific adaptations of the PCAT, item refinement and testing should be guided by formal face and content validity assessments, expert consultation, and engagement with patients and providers . Item refinement should include reviewing relevance of items which may not be relevant to how general practice is currently delivered in Australia (such as after-hours care, home visits, general practice involvement in community health services, discussion of home safety, and facility-related conflict). Consideration should also be given to how provider and specialist services operate within the Australian healthcare system. Instrument development should use data collected specifically for psychometric analysis and aim to sample more widely across demographic groups. More advanced psychometric testing, including assessments of convergent and discriminant validity and exploration of alternative factor structures, would further support the development of a robust and contextually appropriate instrument.

A validated version of the PCAT-S tailored to the Australian context would enhance the ability to measure and compare primary care performance both nationally and internationally, and support ongoing improvement in patient care.”

---

## [Decision Letter · Decision Letter 1]

9 Oct 2025

Dear Dr. Bui,

Thank you for submitting your manuscript to PLOS ONE. After careful consideration, we feel that it has merit but does not fully meet PLOS ONE’s publication criteria as it currently stands. Therefore, we invite you to submit a revised version of the manuscript that addresses the points raised during the review process.

We look forward to receiving your revised manuscript.

Kind regards,

Bojana Bukurov, M.D., Ph.D.

Academic Editor

PLOS ONE

Journal Requirements:

Reviewers' comments:

Reviewer's Responses to Questions

**Comments to the Author**

Reviewer #1: (No Response)

Reviewer #2: (No Response)

2. Is the manuscript technically sound, and do the data support the conclusions?

Reviewer #1: Yes

Reviewer #2: Yes

3. Has the statistical analysis been performed appropriately and rigorously?

Reviewer #1: Yes

Reviewer #2: Yes

4. Have the authors made all data underlying the findings in their manuscript fully available?

Reviewer #1: No

Reviewer #2: Yes

5. Is the manuscript presented in an intelligible fashion and written in standard English?

Reviewer #1: Yes

Reviewer #2: Yes

Reviewer #1: The authors have really made an effort in addressing and responding to all comments. In the introduction, it is clear which references to look at for scale development and validation studies. Nice summary of results, if I understand your statement correctly, the scale could be used in Australian general practice however “results should be interpreted with nuance, and consideration of item relevance and interpretation in the Australian context”. It is clearly stated what further refinements and analyses are needed. However, I do have some additional questions.

I do agree with the authors that temporal stability is important for longitudinal research. However, I am still not convinced that these analyses address test-retest reliability. The idea with test-retest analysis is that the latent variable is assumed to be stable between measurement occasions, thus differences found between measurements indicate errors in the scale. High test-retest reliability indicates that these errors are negligible. In the present study, why is it assumed that the latent variable should be stable? In longitudinal research, sensitivity to change is perhaps even more important than temporal stability. Could these results imply that the instrument is sensitive enough to detect changes?

When looking at table 2, it seems there are high proportions of responses in the highest possible score. The set ceiling effect of above 80% is quite high, how did authors arrive at this level for indicating ceiling effect? Authors have made it clear how this could have impacted the results, and the selected estimator DWLS is appropriate for cases where ceiling effects are high, however I am unfamiliar with the above 80% rule.

Please review the use of periods and commas throughout the text, and typos, for example:

Page 3, line 54, “… and health care providers. and is …”

Page 6, line 139: “… effects were defined as items wwhere > 80% of ...”

Page 20, line 316, “… and to provide a comparison across the three imputation samples.Results …”

Reviewer #2: The commnts from both reviewers were addressed within the revised manuscript. Three concerns remain although I recommend publication without requiring a third round of review. (Note the line numbers refer to the markup version.)

1. The authors should consider citing a source on statistical methods to support the rationale for "conducting an EFA following CFA" (lines 259-260).

2. I echo Reviewer 1's concerns regarding test-retest reliability. I appreicate few studies have evaluated this aspect of the instrument (lines 301-303). The authors note a single test-retest including only 15 respondents, which could indicate the need for further validation (test-restest reliability) of the scale. Still, even that small analysis was conducted over a two-week timeframe rather than a year. Given the low test-retest reliability for the First Contact-Utilization subscale, how can this establish temporal stabliity in a way that's useful for future longitudinal research? No one study can "do it all." If the design of the larger study was such that test-restest reliability was unable to be assessed, I suggest dropping the test-retests analyses form this manuscript.

3. A careful proofreading (and perhaps professional copyediting?) is strongly recommended. Some redudant (or nearly-redundant) language was introduced by the revisions. One example of this near-reptition occurs in lines 655-656: "...reinforcing the PACT's value as a tool for this purpose. This highlights the value of the PACT as a potential tool for this purpose."

**Do you want your identity to be public for this peer review?** For information about this choice, including consent withdrawal, please see our Privacy Policy

Reviewer #1: **Yes:** Maria Fogelkvist

Reviewer #2: **Yes:** Susan L. Schoppelrey

---

## [Author Response · Author response to Decision Letter 2]

24 Nov 2025

Reviewer #1:

I do agree with the authors that temporal stability is important for longitudinal research. However, I am still not convinced that these analyses address test-retest reliability. The idea with test-retest analysis is that the latent variable is assumed to be stable between measurement occasions, thus differences found between measurements indicate errors in the scale. High test-retest reliability indicates that these errors are negligible. In the present study, why is it assumed that the latent variable should be stable? In longitudinal research, sensitivity to change is perhaps even more important than temporal stability. Could these results imply that the instrument is sensitive enough to detect changes?

Author’s response. We thank the reviewers for their comments regarding test-retest reliability. After careful consideration, we have removed the test-retest analyses from the manuscript. We agree that in longitudinal research, the ability of an instrument to detect real changes over time can be more relevant than assuming complete stability of the underlying construct. In our study, the one-year interval between measurements means that observed differences may reflect true changes in participants’ experiences rather than measurement error. While we had previously justified that using a long interval can provide information on temporal stability, this is only in a general sense, and they cannot be interpreted as classical test-retest reliability, which assumes the underlying construct remains stable.

When looking at table 2, it seems there are high proportions of responses in the highest possible score. The set ceiling effect of above 80% is quite high, how did authors arrive at this level for indicating ceiling effect? Authors have made it clear how this could have impacted the results, and the selected estimator DWLS is appropriate for cases where ceiling effects are high, however I am unfamiliar with the above 80% rule.

Author’s response: We thank the reviewer for this comment. Our initial 80% threshold for defining a ceiling effect was based on a previous PCAT validation study (Hoa et al., 2018), but we recognize that this exceeds commonly recommended standards, which typically define floor or ceiling effects as present when more than 15–20% of respondents achieve the minimum or maximum score. We have revised the manuscript to use the 20% threshold, and applying this shows that all but one of the 28 items exhibit high floor or ceiling effects. The results and discussion have been updated accordingly.

Page 6, line 152: “80%” was changed to “20%”

Page 13, lines 318 to 320: “Based on the standard 20% criterion, 5 of the 28 items showed large floor effects and 23 items showed large ceiling effects. Additionally, very high ceiling effects (>80%) were noted for the two first-contact utilization items (C1 and C2) and for item J1.”

Page 26, lines 564 to 565: “In addition, nearly all items exhibited floor or ceiling effects, likely reflecting the homogeneity of our sample.”

Hoa NT, Tam NM, Peersman W, Derese A, Markuns JF. Development and validation of the Vietnamese primary care assessment tool. PLoS One. 2018;13(1):e0191181. Epub 20180111. doi: 10.1371/journal.pone.0191181. pmid: 29324851; PubMed Central PMCID: PMC5764365.

Please review the use of periods and commas throughout the text, and typos, for example:

Page 3, line 54, “… and health care providers. and is …”

Page 6, line 139: “… effects were defined as items wwhere > 80% of ...”

Page 20, line 316, “… and to provide a comparison across the three imputation samples.Results …”

Author response: We thank the reviewer for highlighting these issues. We have carefully proofread the final version of the manuscript to correct any typos.

Reviewer #2: The commnts from both reviewers were addressed within the revised manuscript. Three concerns remain although I recommend publication without requiring a third round of review. (Note the line numbers refer to the markup version.)

Authors’ response: We have added citations to support the rationale for conducting EFA following CFA. First, we cite Flora and Flake (2017, pages 20–21, Chapter “Model Modification”), which discusses moving from a poorly fitting CFA to an exploratory factor analysis to identify alternative factor structures. Second, we cite a practical example of this approach: Ryan et al. (2019) applied EFA after CFA when the hypothesized CFA model did not adequately fit their sample.

Page 8, lines 192 to 194: “ When a hypothesized CFA model does not adequately fit the data, it is acceptable practice to use EFA to explore the underlying factor structure and identify alternative item loadings or configurations [35, 36].”

Flora D, Flake J. The Purpose and Practice of Exploratory and Confirmatory Factor Analysis in Psychological Research: Decisions for Scale Development and Validation. Canadian Journal of Behavioural Science . 2017;49:78–88. doi:10.1037/cbs0000069.

Ryan J, Curtis R, Olds T, Edney S, Vandelanotte C, Plotnikoff R, et al. Psychometric properties of the PERMA Profiler for measuring wellbeing in Australian adults. PLoS One. 2019;14(12):e0225932. doi:10.1371/journal.pone.0225932. PMCID: PMC6927648.

2. I echo Reviewer 1's concerns regarding test-retest reliability. I appreicate few studies have evaluated this aspect of the instrument (lines 301-303). The authors note a single test-retest including only 15 respondents, which could indicate the need for further validation (test-restest reliability) of the scale. Still, even that small analysis was conducted over a two-week timeframe rather than a year. Given the low test-retest reliability for the First Contact-Utilization subscale, how can this establish temporal stabliity in a way that's useful for future longitudinal research? No one study can "do it all." If the design of the larger study was such that test-restest reliability was unable to be assessed, I suggest dropping the test-retests analyses form this manuscript.

Author’s response: We thank the reviewer for their thoughtful comments regarding test-retest reliability. We agree that the design of the larger RCT, which uses a one-year interval, precluded a meaningful assessment of test-retest reliability. After careful consideration, we have therefore removed the test-retest analyses from this manuscript and revised the text accordingly.

3. A careful proofreading (and perhaps professional copyediting?) is strongly recommended. Some redudant (or nearly-redundant) language was introduced by the revisions. One example of this near-reptition occurs in lines 655-656: "...reinforcing the PACT's value as a tool for this purpose. This highlights the value of the PACT as a potential tool for this purpose."

Author’s response: We thank the reviewer for highlighting these issues. We apologize for the oversights and have carefully proofread the final version of the manuscript to remove redundant or repetitive language.

---

## [Decision Letter · Decision Letter 2]

5 Jan 2026

Psychometric properties of the Adult Primary Care Assessment Tool Short form (PCAT-S) among high-risk patients in Australian general practice

PONE-D-24-43966R2

Dear Dr. Chau Minh Bui,

We’re pleased to inform you that your manuscript has been judged scientifically suitable for publication and will be formally accepted for publication once it meets all outstanding technical requirements.

Kind regards,

Bojana Bukurov, M.D., Ph.D.

Academic Editor

PLOS One

Additional Editor Comments (optional):

Reviewers' comments:

Reviewer's Responses to Questions

**Comments to the Author**

Reviewer #1: All comments have been addressed

Reviewer #2: All comments have been addressed

2. Is the manuscript technically sound, and do the data support the conclusions?

Reviewer #1: Yes

Reviewer #2: Yes

3. Has the statistical analysis been performed appropriately and rigorously?

Reviewer #1: Yes

Reviewer #2: Yes

4. Have the authors made all data underlying the findings in their manuscript fully available?

Reviewer #1: No

Reviewer #2: Yes

5. Is the manuscript presented in an intelligible fashion and written in standard English?

Reviewer #1: Yes

Reviewer #2: Yes

Reviewer #1: Thank you for addressing all comments , I have no further questions. I recommend publication of the manusript.

Reviewer #2: (No Response)

**Do you want your identity to be public for this peer review?** For information about this choice, including consent withdrawal, please see our Privacy Policy

Reviewer #1: **Yes:** Maria Fogelkvist

Reviewer #2: **Yes:** Susan L. Schoppelrey

---

## [Editor Report · Acceptance letter]

PONE-D-24-43966R2

PLOS One

Dear Dr. Bui,

I'm pleased to inform you that your manuscript has been deemed suitable for publication in PLOS One. Congratulations! Your manuscript is now being handed over to our production team.

Kind regards,

on behalf of

Ass. prof. Bojana Bukurov

Academic Editor

PLOS One